# Joint angle trajectories are Robust to segment length estimation methods in human reaching

**Rachel I. Taitano**[1*], **Valeriya Gritsenko**[1,2]

1 Department of Neuroscience, School of Medicine, West Virginia University, Rockefeller Neuroscience Institute, West Virginia University, Morgantown, United States of America, 2 Department of Human Performance, Division of Physical Therapy, School of Medicine, West Virginia University, Morgantown, United States of America

* rit0001@mix.wvu.edu

## Abstract

### Background

Quantitative movement analysis is increasingly used to assess motor deficits, but joint angle calculations depend on assumptions about limb segment lengths. These lengths are often estimated from average anthropometric proportions rather than measured directly. The extent to which such assumptions influence joint angle accuracy and variability remains unclear.

### Methods

In prior studies, we recorded reaching movements in nine healthy adults using active-marker motion capture system. In this study, we computed arm joint angles with a dynamic model scaled using either measured segment lengths (Individual method) or proportions based on body height (Average method). Each participant served as their own control across two modeling conditions. We compared segment proportions and the variability in joint angle trajectories arising from segment length assumptions (between-participant variability) with within-participant variability across repeated movements.

### Results

Segment length proportions remained unchanged despite increases in population height. Joint angle trajectories derived from the two scaling methods were very similar. Segment length assumptions had only minor effects on joint angle amplitudes, primarily due to kinematic redundancy, and these effects were substantially smaller than the within-participant variability observed across repeated movements in most individuals. Importantly, while segment length estimates shifted absolute joint angle amplitudes, they did not alter the shape of angular trajectories.

**Data availability statement:** The raw data, such as average time-normalized angular trajectories for individual movements for all degrees of freedom, are shared online on Figshare: (https://doi.org/10.6084/m9.figshare.26885002.v1).

**Funding:** V.G. was supported by National Institute of General Medical Sciences (NIGMS) grants P20GM109098 and P30GM103503. R.I.T. was supported by a fellowship from NIGMS T32 AG052375. This material is based upon work supported by the Office of the Under Secretary of Defense for Research and Engineering under award number FA9550-24-1-0214. Any opinions, findings, conclusions, or recommendations expressed in this material are those of the authors and do not necessarily reflect the views of the U.S. Department of Defense. The funders had no role in study design, data collection and analysis, decision to publish, or preparation of the manuscript.

**Competing interests:** The authors have declared that no competing interests exist.

## Conclusions/significance

Morphological variability in segment lengths contributes less to joint angle variability than the variability expressed by individuals across repeated movements. This indicates that movement variability inherent in movement execution outweighs that introduced by morphological differences. These findings suggest that motion capture–based assessments of reaching quality remain accurate even when segment lengths are inferred from height, supporting their practical use in remote or telehealth clinical assessments where direct anthropometry is not feasible.

## Introduction

In the United States (US), 1 in 4 adults (61 million) live with a disability [1]. The majority of disabilities are often linked to deficits in movement, which can be effectively evaluated through movement analysis. Movement analysis commonly relies on practitioner exploration of active or passive range of motion of specific joints. For more complex movements, subjective observation or rating scales are utilized. For example, the Fugle-Meyer post-stroke impairment assessment includes asking a patient to perform a set of movements, which are rated on a 3-point scale, and the scores are added up [2]. The anthropometric scaling based on body height is widely applied in ergonomic design and workplace safety, where estimations of limb segment lengths from overall height are used to inform equipment and workstation dimensions [3–5]. More recently, metrics based on inverse kinematics, which measure joint angles from tracked body segments, are gaining interest as they help to reduce inter-rater variability and standardize movement analysis [6]. Standardized height-based proportions are widely used to scale kinematic and musculoskeletal models in both marker-based and markerless systems (e.g., OpenSim, Kinect, OpenPose) [7,8]. This approach is especially valuable in remote clinical assessments and tele-rehabilitation, where obtaining direct anthropometric measurements is impractical [6,9–12]. However, it remains unclear how assumptions about segment length affect the accuracy of joint angle estimates derived from such systems.

Movement arises from a complex interaction of neural control, biomechanics, and external environmental factors. Due to the multiple degrees of freedom (DOFs) in our multijointed limbs, the same movement can be performed using different joint angle combinations, each requiring distinct muscle forces. This redundancy complicates clinical assessment, as mildly to moderately impaired patients may produce movements that appear indistinguishable from normal ones [13]. To reduce false positives and inter-rater variability, clinical movement analysis often relies on low-resolution scoring systems that detect only the most apparent movement deficits. However, this approach limits the responsiveness and predictive validity of clinical tests and introduces ceiling effects, particularly in patients with mild motor impairments [14,15].

Redundancy is also an important concept in our modern understanding of how the central nervous system controls movement. Optimal control theory, for example, treats redundancy not as a problem to solve, but as a feature to exploit. The motor

system is hypothesized to exploit redundant DOFs to optimize energy, accuracy, smoothness, or stability under uncertainty, turning infinite solutions into those that best serve behavioral goals [16–19]. Redundancy also has implications in medicine, as appropriately defining and measuring normal redundancy is important for the ability to define and quantify movement deficits.

We define redundancy in arm reaching as variability in the angular trajectories of joint degrees of freedom (DOFs). Within-participant redundancy arises because the multisegmented limb can achieve the same endpoint trajectory through different combinations of joint angles. Between-participant redundancy reflects, in this study, differences in body size, where variations in height are expressed as differences in segment lengths. In our studies, we used virtual reality to elicit standardized reaching movements in healthy adults [20]. By scaling each participant's starting posture and movement amplitudes to their height, we minimize variability in joint angles. This approach yields low-noise normative data on redundancy, that can serve as a reference for future studies on how aging and stroke affect motor control.

Segment lengths between joints are required to process motion capture data and accurately calculate joint angles. When these measures are not obtained directly, they are often estimated from known proportions between individual height and segment lengths [4]. This morphometry has been derived from large cohorts of primarily young adults, mostly male. For example, one of these studies recruited 39 male fighter pilots into three cohorts divided based on volumetric body types (thin, muscular, and rotund) [21]. Based on this and other classical studies, the mean segment length proportions to height were summarized by Winters and others [4,22,23]; these average proportions are widely used to scale biomechanical models [7]. However, the overall height of humans have increased over the course of the XX century due to improvements in nutrition, sanitation, healthcare, and overall living standards [24]. To make sure that the segment proportions generalize to the current taller population, we also included height and segment lengths data from a more recent study of 445 male and 401 female sportspersons [25]. Although the between-participant variability in segment lengths is known from these studies, how it influences the evaluation of between-participant variability in joint angles, and thus the assessment of normative performance, is unknown.

Individual morphological variability in segment lengths influences joint angle amplitudes and the estimates of normal redundancy. Moreover, different methods of measuring or inferring segment lengths from height further influence joint angle calculations. Segment lengths define the kinematic chain of the limb, and thus, joint angle amplitudes. For example, two people whose arms are in the same posture with the same joint angles, but whose segment lengths differ by 10%, have their hands in distinct positions in space (Fig 1A). Conversely if they reach toward the same location in space, their joint angles are different (Fig 1B). Precisely measuring individual segment lengths is often impractical, especially during remote assessments. Therefore, understanding how inference of segment lengths and normative variability in individual segment length proportions affect joint angle calculations is essential for determining the accuracy of assessment algorithms designed to detect movement deficits. This is the goal of the present study. We compared joint angle calculation accuracy using arm segment lengths derived from average body proportions versus direct measurements relative to the typical within- and between-participant variability in joint angles observed in repeated reaching movements.

## Methods

### Participants and experimental design

The Institutional Review Board of West Virginia University approved the experimental protocol (Protocol #1311129283). Volunteers were recruited through fliers distributed around Morgantown, West Virgina. The data collection started on March 12th, 2014, and ended on July 25th, 2016. Volunteers provided their written informed consent prior to the start of the experiment. The signing of the informed consent form was witnessed by a second member of the research team.

Nine healthy human volunteers (22.8 +/- 0.67 years, 6M:3F) performed a center-out reaching task comprising reaching toward visual targets presented in virtual reality as described in detail in Olesh et al. [20]. The use of virtual reality was convenient as it helped standardize the instructions to participants and provided an engaging method to maintain

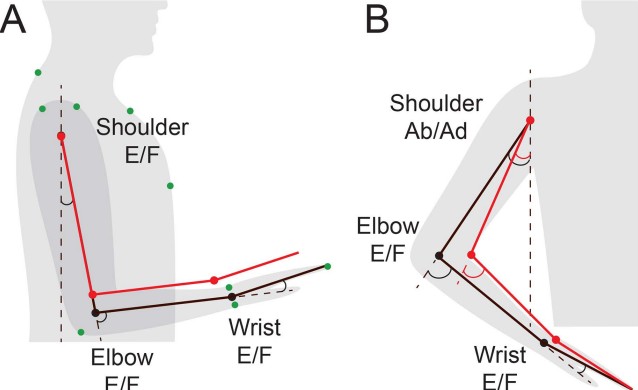

**Fig 1. Illustration of kinematics.** Red segments are 10% shorter each than the black segments. Dashed lines show axes against which the joint angles were measured. E/F stands for extension/flexion degree of freedom of the corresponding joint. A. When the joint angles are kept the same, the changes in limb segment lengths change the location where the hand can reach. Green dots illustrate the locations of markers used for motion capture. B. When the tip of the hand is kept the same, the changes in limb segment lengths change joint angles.

their attention during the data collection session. An Oculus Rift virtual reality headset was used to display visual targets as spheres of 2 cm diameter. The targets appeared sequentially at the center location, corresponding to an arm position with a shoulder flexion and abduction at 0° with the upper arm parallel to the trunk, elbow flexion of 90° with the forearm parallel to the floor, and wrist flexion of 0° with palm down (hand pronated), and at one of 14 locations along a sphere equidistant from a center location, followed by return to the center location. The order of the 14 locations was randomized. Participants were instructed to point with their index finger to each of the targets as quickly and as accurately as they can without flexing their wrist. Target placement within the virtual task was set based on each participant's arm length to minimize inter-participant variability in starting joint angles and preserve an equal range of motion. Each reaching movement from the center out and return was repeated 15 times.

Reaching movements were recorded using an active-marker Impulse system (PhaseSpace). The light-emitting diode (LED) markers were placed on bony landmarks according to standard practices [26] (Fig 1A). We selected the PhaseSpace Impulse active-marker motion capture system for its high spatial and temporal resolution (submillimeter accuracy at 480 Hz), which is essential for capturing subtle joint kinematics in upper-limb reaching tasks. Unlike passive-marker systems, the active markers used by PhaseSpace reduce marker occlusion and provide robust signal detection, particularly important in experiments involving multiple closely spaced segments such as the upper limb and hand. Additionally, the individual self-reported height was recorded, and the lengths of the humerus, radial/ulnar, and hand segments were measured at the time of the experiment. The length of the humerus segment was measured between the acromion and the lateral epicondyle. The length of the radioulnar segment was measured between the lateral epicondyle and the location between the styloid processes of the ulna and radius. The length of the hand segment was measured between the median of the styloid processes and the tip of the index finger. The index finger was used because the participants were instructed to point with that finger, and its tip was instrumented with the marker. The motion capture data comprising coordinates of 9 LED markers in time and the participant's height and segment lengths were imported into MATLAB (Mathworks) for analysis. The coordinates were recorded at 480 Hz temporal resolution and submillimeter spatial resolution.

## Data analysis

A dynamic model of the arm and hand with 27 degrees of freedom and 20 segments was developed using the Simscape Multibody toolbox in MATLAB and used to obtain joint angles as described in detail in Bahdasariants et al. [27]. Most



segments were modelled as cylinders except the trunk and metacarpophalanges, which were modelled as rectangular prisms. The lengths of the humeral and radioulnar segments and the overall length of the hand (hand segment) were changed in this study. The hand length comprised the lengths of the metacarpophalanges plus three phalanges of the index finger. We compared the effects on joint angles of the two methods for scaling the arm segments. The first method, termed here the Average method, relied on published anthropomorphic dimensions as fractions of individual height, namely 0.1880 for the humeral segment length, 0.1450 for the radioulnar segment length, and 0.1080 for the hand length [4,28]. The second method, termed here the Individual method, used the segment lengths measured directly from individuals as described above.

Next, the trajectories of marker coordinates were low-pass filtered at 10 Hz and interpolated with a cubic spline. Then, simulations were run in the Simulink toolbox of MATLAB using marker trajectories for each recorded movement to drive the movement of virtual markers attached to model segments through linear springs. The resulting angular trajectories of three shoulder DOFs (flexion/extension, abduction/adduction, and internal/external rotation), one elbow DOF (flexion/extension), one radioulnar DOF (pronation/supination), and two wrist DOFs (flexion/extension and radial/ulnar deviation) were recorded for each segment scaling method per reaching movement and per participant. For subsequent analyses, all joint angle trajectories were first normalized to the duration between the start and end of each movement and then resampled at 100 samples. The movement ended once participants returned to the central position (Fig 3, insert).

The angular trajectories calculated with the Individual method were used to quantify the active range of motion. The time-normalized trajectories were averaged per reaching direction and minimal values across all trajectories were subtracted from the maximal values. The maximal range of motion was estimated from an ergonomics textbook [5].

The segment lengths of individuals recruited into our small-cohort study were compared to two previous studies. The earlier study completed in 1955 is widely used as basis for segment scaling based on individual height [21]. A more recent study from 2009 recruited taller individuals [25] whose heights were more comparable to our cohort (Table 1). However, the segment proportions across the two prior studies were similar. Therefore, we used the most widely used proportions from the earlier study [4] here as part of the Average method for estimating between-participant variability.

## Statistics and reproducibility

Statistical analysis compared the within- and between-participant variability. The within-participant variability ($\sigma_w$) for a given reaching movement was calculated as follows:

$$\sigma_w = \frac{1}{100} \sum_{t=1}^{100} \sqrt{\frac{1}{N} \sum_{i=1}^{N} \left( \theta_i(t) - \bar{\theta}(t) \right)^2},$$

**Table 1. Segment lengths calculated with the two methods and data from two reference studies.**

| Length measurements | From Dempster et al. 1955 (cm) | From Canda 2009 (cm) | Individual method (cm) | Average method (cm) | Difference (cm) |
|---|---|---|---|---|---|
| Humerus | 35.3 ± 1.7 | M: 33.9 ± 2.1<br>F: 31.3 ± 1.9 | 35 ± 2.2 | 34 ± 1.2 | 1.6 ± 1.0 |
| Radius/ulnar | 27.3 ± 1.1 | M: 26.2 ± 1.7<br>F: 23.7 ± 1.5 | 26 ± 2.1 | 26 ± 0.9 | 1.3 ± 1.3 |
| Hand | 19.1 ± 1.0 | M: 19.6 ± 1.1<br>F: 17.9 ± 0.9 | 15 ± 1.1 | 16 ± 0.6 | 0.1 ± 0.1 |
| Height | 175.1 ± 4.3 | M: 179.4 ± 9.1<br>F: 166.5 ± 7.6 | 180 ± 6.4 | | |

Values are means ± standard deviation across individuals; the Difference was calculated between lengths measured with Individual and Average methods.



where $N$ is the number of repetitions of the reaching movements to a given target (max 15), $\theta_i$ is a given DOF's angle during $i$th reach to a given target from simulations with individually measured segments (Fig 2, red shaded areas), $\overline{\theta}$ is the mean across trajectories, and t is normalized time in samples (total t = 100).

The between-participant variability was calculated as the standard deviation ($\sigma_b$) between the corresponding angular trajectories from simulations with averaged and individual segments as follows:

$$\sigma_b = \frac{1}{100} \sum_{t=1}^{100} \sqrt{\frac{1}{N} \sum_{i=1}^{N} (\varnothing_i(t) - \theta_i(t))^2},$$

where $N$ is the number of repetitions of the reaching movements to a given target (max 15), $\theta_i$ is a given DOF's angle during $i$th reach to a given target from simulations with individually measured segments (Fig 2, red shaded areas), $\varnothing_i$ in the corresponding DOF angle from simulations with averaged segments (Fig 2, blue shaded areas), and t is normalized time in samples (total t = 100). In this analysis, the angles estimated from averaged segment dimensions act as a stand-in for a hypothetical participant of the same overall height but with different proportions of limb segments. However, this underestimates true between-participant variability. The actual variability—calculated by comparing each participant's mean trajectory with the overall group mean—would be larger, since it would also capture differences in total body height across participants.

To determine whether the discrepancies in joint angles due to segment length measurements are larger than the normal variability of motion, repeated-measures analysis of variance (ANOVA) was performed on differences $\sigma_w - \sigma_b$ in degrees using *fitrm* function. The differences were normally distributed based on the Kolmogorov-Smirnov test applied using *kstest* function. The participant's sex assigned at birth served as a between-participant factor; the within-participant factors were movement direction (Target # 1:14, Fig 4 insert) and DOF (7 DOFs of major arm and forearm joints listed above).

We have also calculated effect size using Cohen's d, which quantifies the standardized difference between two means for paired (within-subject) comparisons calculated as:

$$d = \frac{\overline{\sigma}_w - \overline{\sigma}_b}{s_d},$$

where $\overline{\sigma}_w$ and $\overline{\sigma}_b$ are the mean within and between variability, and $s_d$ is the standard deviation of the difference between them.

To evaluate whether differences in joint angle trajectories due to segment length estimation methods were localized in time, we performed a one-dimensional statistical parametric mapping (1D-SPM) analysis on joint angle time series. Mean time-normalized trajectories were stacked in time across the movements towards 14 target locations and combined across subjects to create a matrix of 9 x 14000 per DOF. Paired t-tests were conducted using the open-source SPM1D v0.4 toolbox for MATLAB [29] to compare joint angle profiles between the segment-length estimation methods. The t-statistics were computed over the entire trajectory, and significance was determined using random field theory–based thresholds at α = 0.05, corrected for the 1D nature of the data. The analysis was performed separately for each joint angle using the function spm1d.stats.ttest_paired, and the resulting supra-threshold clusters were visualized for interpretation.

## Results

Here, we investigated how the different methods of inferring segment lengths affect the amplitude of joint angles during reaching movements. The segment lengths of individuals recruited into our study were comparable to those from the recent study with comparable individual heights [25] (Table 1). The segment length proportions derived from taller individuals were 0.1899 & 0.1879 for the humeral segment length of males & females respectively, 0.1460 & 0.1423 for the



**Fig 2. Example of the differences between joint angle trajectories calculated with models scaled in two ways for a single movement in one direction (and back) performed by one participant.** Joint angle trajectories were calculated from musculoskeletal models scaled using average proportions (black) and individual arm segment lengths (red). Thick lines show average trajectories and shaded areas show standard deviation across 15 repetitions for the same movement. Abbreviations: E/F = extension/flexion; Ab/Ad = abduction/adduction; E/I = external/internal; R/U = radial/ulnar.

radioulnar segment length of males & females respectively, and 0.1093 & 0.1075 for the hand length of males & females respectively. These proportions are similar to those derived from shorter individuals and used in our study (0.186,0.146, and 0.108 for humeral, radioulnar, and hand segment length respectively) [4]. This shows that, on average, the segment proportion are very similar between short and tall individuals. Table 1 further shows that the segment lengths estimated

from height were less variable than those measured directly (standard deviation for the Average compared to the Individual method). However, the variability in the measurement of individual segment lengths fell within the natural variability between individuals, thus, it is unlikely to limit the validity of the study conclusions.

The ranges of motion observed in major arm DOFs during the recorded reaching movements comprised a fraction of the joints' anatomical maximum range of motion (Table 2). Therefore, it is important to interpret the differences in angles calculated with different segment lengths relative to the joints' ranges of motion. Our analysis has shown that the joint angle trajectories calculated using models scaled with the Average and Individual methods were very similar (Fig 2). One-dimensional SPM analysis revealed no significant differences between segment length estimation methods for all DOFs (all p > 0.05; Fig 3). This is because the marker coordinates constrain and define the temporal profiles of the joint angles. However, for some DOFs in some individuals, the initial and starting angles were unequal (Fig 2, shoulder abduction/adduction and rotation DOFs). This is because of the redundancy of multijointed limbs, as illustrated in Fig 1B. These offsets drove the $\sigma_b$ values. However, they were still a small fraction of the active ranges of motion (Table 2, last column) and were much smaller than $\sigma_w$ in most participants (Fig 4).

Repeated-measures ANOVA also found that $\sigma_b$ was consistently smaller than $\sigma_w$ (Table 3, Intercept). The values did not differ significantly between males and females (Table 3, Sex). However, values of $\sigma_w - \sigma_b$ differed systematically across targets and DOFs (Table 3, Intercept:Target, Intercept:DOF). There was no evidence that Sex modulated the Target nor DOF effects (Table 3, Sex:Target, Sex:DOF), but the amplitudes of $\sigma_w - \sigma_b$ across targets varied across degrees of

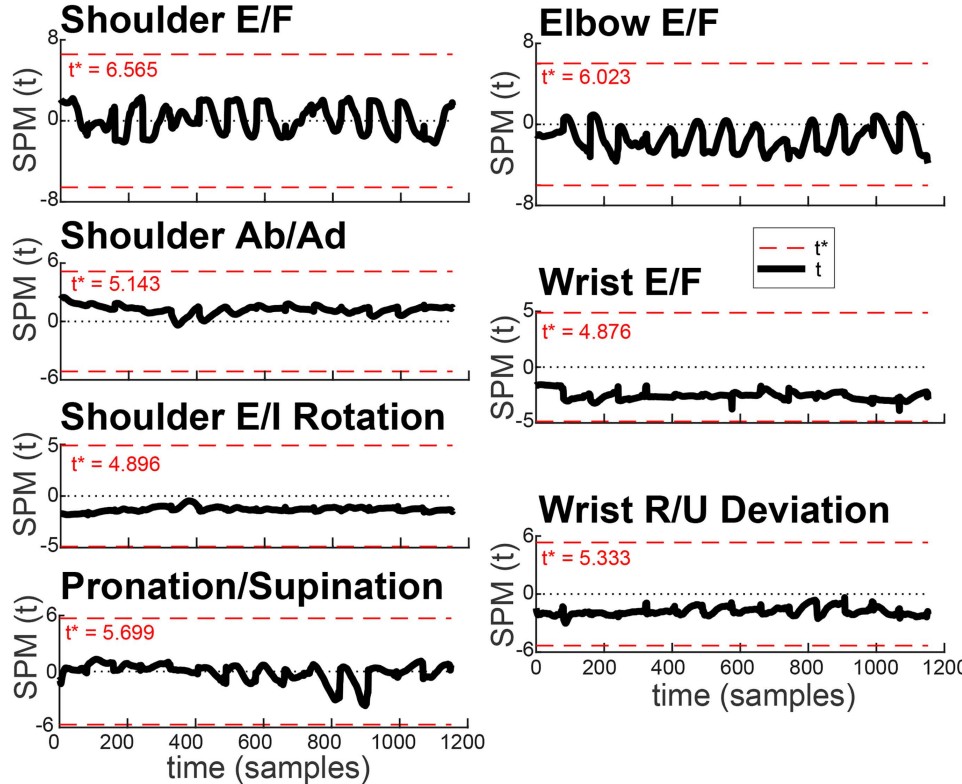

**Fig 3. Statistical parametric mapping analysis of joint angle time series comparing segment length estimation methods.** The black curve represents the one-dimensional statistical parametric mapping (SPM) t-statistic as a function of time. The red dashed line indicates the critical threshold (t*) based on random field theory at α = 0.05. No supra-threshold clusters were identified. Abbreviations: E/F = extension/flexion; Ab/Ad = abduction/adduction; E/I = external/internal; R/U = radial/ulnar.

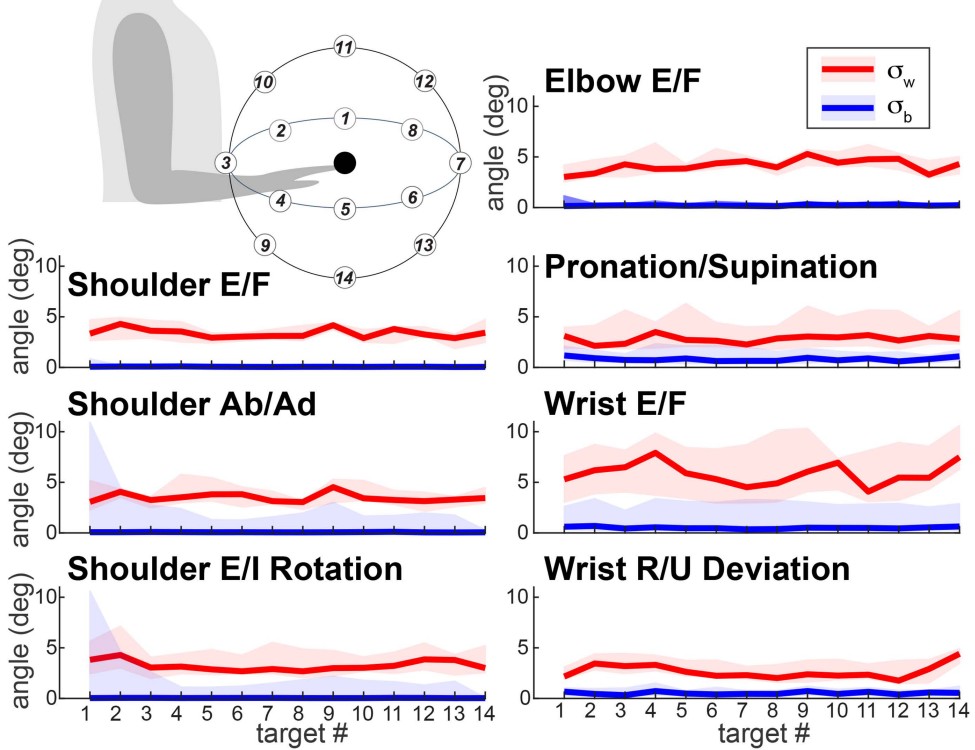

**Fig 4. Comparison between within- and between-participant variability.** Values of between-participant variability ($\sigma_b$ blue) and within-participant variability ($\sigma_w$) were averaged across all participants (lines indicate mean values and shaded areas indicate standard deviation across participants) for each target location and degree of freedom. The insert shows the locations of targets 1-14 relative to the participant's position (drawn not to scale). Abbreviations: E/F = extension/flexion; Ab/Ad = abduction/adduction; E/I = external/internal; R/U = radial/ulnar.

**Table 2. Angle errors between scaling methods relative to the total joint range of motion.**

| Joint DOF | Active ROM (deg) | Maximal ROM (deg) | 25%−75% range of $\sigma_b$ (deg) | 25%−75% range of $\sigma_w$ (deg) |
|---|---|---|---|---|
| Shoulder flexion/extension | 32 | 220 | 0.1–0.2 | 2.3–4.0 |
| Shoulder abduction/adduction | 33 | 180 | 0.1–1.9 | 3.0–6.0 |
| Shoulder internal/external rotation | 25 | 170 | 0.0–1.7 | 2.8–7.6 |
| Elbow flexion/extension | 57 | 130 | 0.1–0.4 | 3.3–5.1 |
| Supination/pronation | 17 | 160 | 0.5–1.7 | 1.9–5.3 |
| Wrist flexion/extension | 21 | 150 | 0.3–3.0 | 3.3–7.0 |
| Wrist radial/ulnar deviation | 13 | 50 | 0.3–1.0 | 2.2–4.0 |

ROM stands for range of motion; $\sigma_b$ stands for between-participant standard deviation; $\sigma_w$ stands for within-participant standard deviation.

freedom (Table 3, Intercept:Target:DOF). There was no significant three-way interaction (Table 3, Sex:Target:DOF). We estimated a post hoc effect size (Cohen's d) for the primary comparison between joint angle variability from the Average versus Individual segment scaling methods. Across joints, the difference in variability ($\sigma_w − \sigma_b$) ranged from ~1.5° to 4.0°, with a conservative estimate of standard deviation of differences of ~1.5°. This yielded an estimated Cohen's d between 1.5 and 2.5, consistent with a very large effect size, and a post hoc power > 0.9 at α = 0.05. This supports the adequacy of



**Table 3. Repeated-measures ANOVA results.**

|  | Sum of Squares | DF | F-statistic | *p-value* |
|---|---|---|---|---|
| Intercept | 2.17 | 1 | 28.16 | **0.001** |
| Sex | 0.03 | 1 | 0.38 | 0.55 |
| Intercept:Target | 1.66 | 1 | 23.52 | **0.002** |
| Sex:Target | 0.05 | 1 | 0.72 | 0.43 |
| Intercept:DOF | 1.76 | 1 | 28.73 | **0.001** |
| Sex:DOF | 0.02 | 1 | 0.29 | 0.61 |
| Intercept:Target:DOF | 1.35 | 1 | 24.54 | **0.002** |
| Sex:Target:DOF | 0.03 | 1 | 0.59 | 0.47 |

Intercept refers to the baseline level of the dependent variable when all factors are set to zero; a semicolon shows interaction effects between factors; DF is degrees of freedom.

our sample size for determining that the differences between segment scaling methods have a lower impact on joint angle calculations than the normal within-participant variability due to the redundancy of the multijointed limb. These findings show that joint angle trajectories remain robust when segment lengths are inferred from height, supporting the use of height-scaled models in telehealth-based assessments and motion capture tools where segment measurements are not practical.

## Discussion

In this study, we estimated joint angles from motion capture during reaching movements and examined how assumptions about segment length affect joint angle trajectories. We found that despite the increased height of the population, the segment proportions stayed the same on average. We also found that segment length assumptions had minimal influence on forearm joint angles for most individuals (Fig 3, 4). The small variations that did occur reflected kinematic chain redundancy, which allows a multijointed limb to achieve the same endpoint trajectory with different joint angle combinations. Importantly, these variations were far smaller than the normal within-participant variability observed in reaching trajectories. Thus, clinical assessments of active or passive range of motion, as well as movement quality, are unlikely to be affected by the method used to estimate segment lengths. By demonstrating that the resulting joint angle trajectories remain robust, even when using generalized anthropometric estimates, our findings support the use of height-based segment scaling in both research and clinical settings, including remote or home-based assessments where direct measurements are not feasible. This has important implications for tele-rehabilitation, automated video-based assessment tools, and scalable screening technologies.

Clinical rating scales for assessing movement deficits offer significantly lower resolution compared to even the lowest-quality motion capture systems [6,9,30,31]. Previously, we found that movement quality assessments are sensitive to the shape of joint angle trajectories [6,13]. In this study, we demonstrated that segment length measurement methods do not affect the shape of joint angle trajectories, even though they do influence overall joint angle amplitudes (Fig 2). Taken together, these findings suggest that the quality of movement assessment is unlikely to be affected by the method of segment length measurement.

Our findings indicate that morphological variability between individuals contributes less to joint angle variability than the variability expressed by individuals across repeated movements. However, there are several limitations to the generalizability of our conclusions. The findings demonstrate the robustness of joint angle calculations to segment scaling assumptions in the context of upper-limb reaching tasks and should not be overgeneralized to lower limbs or all forms of movement, such as fine object manipulation. Other factors, such as neural noise, sensory uncertainty, or inconsistent control strategies, may also contribute to the observed between-participant variability. Moreover, in clinical populations,



neurological or muscular impairments can fundamentally alter motor variability and the temporal structure of movements. Furthermore, for obvious reasons, the method does not apply to congenital, developmental, musculoskeletal, or traumatic conditions that alter the segment length proportions. Therefore, additional validation is needed to confirm whether height-inferred segment scaling remains reliable for motor deficit assessment in various clinical populations.

In summary, our findings show that assumptions about segment length have minimal impact on joint angle trajectories during reaching, with any small variations reflecting kinematic redundancy rather than morphological differences. These variations were far smaller than the natural within-participant variability observed across repeated movements for most individuals, indicating that movement variability is more influenced by movement execution and task redundancy than by small morphological differences. Importantly, segment length estimation methods did not alter the shape of joint angle trajectories, suggesting that clinical assessments of movement quality or range of motion are unlikely to be affected. Together, these results highlight that variability in motor performance appears to be more influenced by task-related factors and movement execution variability than by morphological differences in segment lengths. By demonstrating that joint angle calculations are robust to height-based scaling, this study supports the use of motion capture for scalable, remote, and automated assessment of movement quality, including in telemedicine and rehabilitation contexts where direct anthropometric measurement is not feasible.

## Acknowledgments

We would like to acknowledge the contribution of Mychaela MacDonald to the analysis of the reported data as part of her undergraduate Research Apprenticeship Program at West Virginia University.

## Author contributions

**Conceptualization:** Rachel I. Taitano, Valeriya Gritsenko.

**Data curation:** Rachel I. Taitano, Valeriya Gritsenko.

**Formal analysis:** Rachel I. Taitano, Valeriya Gritsenko.

**Funding acquisition:** Valeriya Gritsenko.

**Investigation:** Valeriya Gritsenko.

**Methodology:** Rachel I. Taitano, Valeriya Gritsenko.

**Project administration:** Valeriya Gritsenko.

**Resources:** Valeriya Gritsenko.

**Software:** Rachel I. Taitano, Valeriya Gritsenko.

**Supervision:** Valeriya Gritsenko.

**Validation:** Valeriya Gritsenko.

**Visualization:** Valeriya Gritsenko.

**Writing – original draft:** Rachel I. Taitano, Valeriya Gritsenko.

**Writing – review & editing:** Rachel I. Taitano, Valeriya Gritsenko.

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
