## [Decision Letter · Decision Letter 0]

17 Oct 2024

PONE-D-24-38117Evaluating Joint Angle Data for Clinical Assessment Using Multidimensional Inverse Kinematics with Average Segment Morphometry.PLOS ONE

Dear Dr. Gritsenko,

Thank you for submitting your manuscript to PLOS ONE. After careful consideration, we feel that it has merit but does not fully meet PLOS ONE’s publication criteria as it currently stands. Therefore, we invite you to submit a revised version of the manuscript that addresses the points raised during the review process.

The reviewers raised several significant issues with the manuscript. Reviewer #1 points out the use of bulk citations (e.g., line 51) and mentions the lack of a detailed statistical analysis despite mentioning the ANOVA method. Reviewer #2 criticizes the manuscript for missing essential details, making it difficult to understand the experiment, which aimed to calculate joint angles using musculoskeletal models with different scaling methods in a virtual reality setup. The abstract lacks clarity on the methods, tasks, and results, while the introduction fails to clearly articulate the study’s contribution. The methods and data analysis are described too compactly, omitting crucial information such as task descriptions, repetitions, and data processing steps. The results section lacks clear explanations for the values presented, and the discussion fails to provide concrete conclusions or explain certain claims. Both reviewers recommend significant revisions to improve readability, clarity, and the inclusion of missing data. The authors are suggested to consider these concerns seriously in the revised version of the manuscript.

We look forward to receiving your revised manuscript.

Kind regards,

Jyotindra Narayan

Academic Editor

PLOS ONE

Journal Requirements:

-Applications and limitations of current markerless motion capture methods for clinical gait biomechanics (https://peerj.com/articles/12995/)

-Advancing Medical Technology for Motor Impairment Rehabilitation: Tools, Protocols, and Devices (https://doi.org/10.33915/etd.12056)

(among others)

In your revision ensure you cite all your sources (including your own works), and quote or rephrase any duplicated text outside the methods section. Further consideration is dependent on these concerns being addressed.

3. Please note that PLOS ONE has spec6ific guidelines on code sharing for submissions in which author-generated code underpins the findings in the manuscript. In these cases, all author-generated code must be made available without restrictions upon publication of the work. Please review our guidelines at https://journals.plos.org/plosone/s/materials-and-software-sharing#loc-sharing-code and ensure that your code is shared in a way that follows best practice and facilitates reproducibility and reuse.

“NIGMS P20GM109098, NIGMS P30GM103503, NIGMS T32 AG052375, Office of the Under Secretary of Defense for Research and Engineering FA9550-24-1-0214”

5. Please expand the acronym “NIGMS” (as indicated in your financial disclosure) so that it states the name of your funders in full.

“We would like to acknowledge the contribution of Mychaela MacDonald to the analysis of the reported data as part of her undergraduate Research Apprenticeship Program at West Virginia University. V.G. was supported by NIGMS grants P20GM109098 and P30GM103503. R.I.T. was supported by a fellowship from NIGMS T32 AG052375. This material is based upon work supported by the Office of the Under Secretary of Defense for Research and Engineering under award number FA9550-24-1-0214. Any opinions, findings and conclusions or recommendations expressed in this material are those of the authors and do not necessarily reflect the views of the U.S. Department of Defense”

“NIGMS P20GM109098, NIGMS P30GM103503, NIGMS T32 AG052375, Office of the Under Secretary of Defense for Research and Engineering FA9550-24-1-0214”

Reviewers' comments:

Reviewer's Responses to Questions

**Comments to the Author**

1. Is the manuscript technically sound, and do the data support the conclusions?

Reviewer #1: Yes

Reviewer #2: No

2. Has the statistical analysis been performed appropriately and rigorously? 

Reviewer #1: I Don't Know

Reviewer #2: I Don't Know

3. Have the authors made all data underlying the findings in their manuscript fully available?

Reviewer #1: Yes

Reviewer #2: No

4. Is the manuscript presented in an intelligible fashion and written in standard English?

Reviewer #1: Yes

Reviewer #2: Yes

5. Review Comments to the Author

Reviewer #1: I noticed in many cases bulk citations that should be avoided, e.g. at line 51 you cite [2-9].

At line 164 you have {Citation}.

The paper does not present the statistical analysis that was made. It is mentioned ANOVA method, but without details.

Reviewer #2: This paper describes an experiment where the goal was to calculate joint angles with a musculoskeletal model that was personalized using an Average method and an Individual method in an experiment that used virtual reality. While I believe that the study could be interesting, at this point the manuscript misses a lot of vital information that is required to understand what was being done. Therefore, I cannot recommend publication of the study at this point.

The text is so compact in the manuscript that it is a bit difficult to read the paper. It would be great if the text is double spaced.

It would also be great if the raw experimental data could be shared as well.

Abstract: 8 out of 20 lines are a rather general introduction, which does not explain why the authors were specifically interested in testing different scaling methods. Furthermore, the methods are not described clearly: which motion capture technology is being tested? What sort of tasks were performed? How many participants? Also, it would be great if some quantitative results were added.

Introduction: while I generally liked the introduction, a statement/final paragraph is missing which describes exactly the contribution of the study that is described in the manuscript. Therefore, it is really difficult to assess the usefulness of the methodology and the results towards the contribution. For example, why did the authors choose the specific tests that they describe in line 146-154.

Line 80: The authors could further expand on the redundancy as it is quite relevant to their study.

Methods:

• Line 114: please describe the type of virtual reality that was being used. Why is the virtual reality relevant?

• Please describe (here or in the introduction) how comparable the experimental setup is to the one that is expected to be used in the clinic

• Please describe the experiment and data processing more clearly: what tasks were measured? How many repetitions? What was done to the measured data exactly?

• I generally did not understand the data analysis section. The authors have described it a bit too densely. For example, it is only described that the “Average method” relies “on published anthropomorphic dimensions as fractions of individual height”, but how were these applied? And which segment lengths were changed in the “Individual method”. Why was a time-independent standard deviation used?

Results: there are more results mentioned in table 1 than just the Average and Individual method. Please explain where these values come from. For table 2, since the tasks are not described, it is difficult to understand what the ROM means.

Discussion:

• Line 197-198: I do not understand this sentence. According to the methods, there were two methods and there is not a separate ground truth, so how can one “alter the values the most”.

• Line 211-212: what is “good enough” based on?

• A concluding paragraph is missing.

6. PLOS authors have the option to publish the peer review history of their article (what does this mean? ). If published, this will include your full peer review and any attached files.

**Do you want your identity to be public for this peer review?** For information about this choice, including consent withdrawal, please see our Privacy Policy .

Reviewer #1: No

Reviewer #2: No

---

## [Author Response · Author response to Decision Letter 1]

21 Feb 2025

Response to reviewers

Our responses to the editor and reviewers are presented in bold writing. All page numbers cited in our responses refer to the ‘clean’, revised manuscript.

Reviewer #1

Q. I noticed in many cases bulk citations that should be avoided, e.g. at line 51 you cite [2-9]. At line 164 you have {Citation}.

A. Thank you for bringing this to our attention. We have revised the corresponding portions of the Introduction and Discussion to elaborate on relevant findings from previous studies and corrected the error.

Q. The paper does not present the statistical analysis that was made. It is mentioned ANOVA method, but without details.

A. The description of ANOVA was in the last paragraph of the Methods section immediately following the description of RMSE, now on lines 163-177. We have added a section title, Statistics and Reproducibility, to highlight it.

Reviewer #2

Q. This paper describes an experiment where the goal was to calculate joint angles with a musculoskeletal model that was personalized using an Average method and an Individual method in an experiment that used virtual reality. While I believe that the study could be interesting, at this point the manuscript misses a lot of vital information that is required to understand what was being done. Therefore, I cannot recommend publication of the study at this point.

A. Thank you for the supportive feedback. We have detailed the changes to the concerns and comments below.

Q. The text is so compact in the manuscript that it is a bit difficult to read the paper. It would be great if the text is double spaced.

A. Thank you for the suggestion. We have changed the line spacing of the main text for your additional review and updated all formatting to match the PLOS One guidelines.

Q. It would also be great if the raw experimental data could be shared as well.

A. We have added the raw experimental data to the Figshare repository. The data includes the joint angle timeseries calculated with the two methods per target per subject per degree of freedom.

Q. Abstract: 8 out of 20 lines are a rather general introduction, which does not explain why the authors were specifically interested in testing different scaling methods. Furthermore, the methods are not described clearly: which motion capture technology is being tested? What sort of tasks were performed? How many participants? Also, it would be great if some quantitative results were added.

A. Thank you for the suggestion. We have added the requested information into the abstract and rewrote it to include more details.

Q. Introduction: while I generally liked the introduction, a statement/final paragraph is missing which describes exactly the contribution of the study that is described in the manuscript. Therefore, it is really difficult to assess the usefulness of the methodology and the results towards the contribution. For example, why did the authors choose the specific tests that they describe in line 146-154.

A. We have included additional clarifications and explicit statements about the goal of the study on lines 90 - 101.

Q. Line 80: The authors could further expand on the redundancy as it is quite relevant to their study.

A. We have followed this suggestion and expanded on redundancy now on lines 60 – 79.

Q. Methods, line 114: please describe the type of virtual reality that was being used. Why is virtual reality relevant?

A. Thank you for the suggestion. We have added more details on lines 110 – 124.

Q. Methods: Please describe (here or in the introduction) how comparable the experimental setup is to the one that is expected to be used in the clinic

A. Thank you for this point. We have expanded on how we envision this technology can be used for future clinical assessment in Discussion on lines 240-252.

Q. Methods: please describe the experiment and data processing more clearly: what tasks were measured? How many repetitions? What was done to the measured data exactly?

A. We have expanded the description in Methods on lines 110-137.

Q. Methods: I generally did not understand the data analysis section. The authors have described it a bit too densely. For example, it is only described that the “Average method” relies “on published anthropomorphic dimensions as fractions of individual height”, but how were these applied? And which segment lengths were changed in the “Individual method”. Why was a time-independent standard deviation used?

A. We have expanded the Data Analysis section in Methods now on lines 140-160.

Q. Results: there are more results mentioned in table 1 than just the Average and Individual method. Please explain where these values come from. For table 2, since the tasks are not described, it is difficult to understand what the ROM means.

A. We have expanded the description of Table 1 in Results on lines 179-192 and of Table 2 on lines 193-208. We have also added the active ROM calculated from the reaching movements in addition to the maximal ROM and added RMSE ranges for each DOF in Table 2.

Q. Discussion, line 197-198: I do not understand this sentence. According to the methods, there were two methods and there is not a separate ground truth, so how can one “alter the values the most”.

A. This sentence was removed, instead, we expanded on the clinical applicability of our findings in the Discussion on lines 221-252.

Q. Discussion, line 211-212: what is “good enough” based on?

A. This statement was removed.

Q. A concluding paragraph is missing.

A. We added a concluding paragraph on lines 253-264.

---

## [Decision Letter · Decision Letter 1]

18 Mar 2025

PONE-D-24-38117R1Quantifying the Effect of Anthropometric Assumptions on Upper Limb Joint Angles Derived from Inverse Kinematics.PLOS ONE

Dear Dr. Gritsenko,

Thank you for submitting your manuscript to PLOS ONE. After careful consideration, we feel that it has merit but does not fully meet PLOS ONE’s publication criteria as it currently stands. Therefore, we invite you to submit a revised version of the manuscript that addresses the points raised during the review process.

Despite the favourable feedabck by the reviewer(s), Reviewer #2 highlights major concerns, particularly regarding the results section and data availability. Key issues include unclear methodology descriptions, inconsistent terminology usage, missing analysis details, and misinterpretations of statistical results. Minor comments focus on readability, terminology refinement, and additional clarifications in the introduction and discussion.

We look forward to receiving your revised manuscript.

Kind regards,

Jyotindra Narayan

Academic Editor

PLOS ONE

Reviewers' comments:

Reviewer's Responses to Questions

**Comments to the Author**

1. If the authors have adequately addressed your comments raised in a previous round of review and you feel that this manuscript is now acceptable for publication, you may indicate that here to bypass the “Comments to the Author” section, enter your conflict of interest statement in the “Confidential to Editor” section, and submit your "Accept" recommendation.

Reviewer #1: All comments have been addressed

Reviewer #2: (No Response)

2. Is the manuscript technically sound, and do the data support the conclusions?

Reviewer #1: Yes

Reviewer #2: Partly

3. Has the statistical analysis been performed appropriately and rigorously? 

Reviewer #1: Yes

Reviewer #2: No

4. Have the authors made all data underlying the findings in their manuscript fully available?

Reviewer #1: Yes

Reviewer #2: No

5. Is the manuscript presented in an intelligible fashion and written in standard English?

Reviewer #1: Yes

Reviewer #2: Yes

6. Review Comments to the Author

Reviewer #1: (No Response)

Reviewer #2: The authors have addressed my previous comments very well, which has greatly improved the manuscript. The purpose of the paper is now clear, and the methodology is much clearer as well. However, several major issues remain, especially in the results section.

I am not sure if the data that is made available meets the standard required by PLOS One, as it is only the root-mean-square errors and no raw data, or perhaps time series data of joint angles over time. E.g., Figure 2 cannot be reproduced as far as I can see.

Major Comments

Line 28-30: This sentence does not describe the purpose of the paper correctly. As far as I understood, the goal of the paper is to investigate if joint angles can be estimated from a model scaled from average body proportions as accurately as when using a model with measured body proportions.

Line 56: the statement “obstacle to widespread adoption” is contradictory to line 51 which claims that there already is widespread adoption.

Throughout (line 94/98/…): it is confusing to me that the word “variability” is used with at least two different meanings, the variability in segment length and the variability in reaching movement execution. Furthermore, it also seems to me that sometimes “variability in segment lengths” is used to describe the fact that there is variation in measurements, not necessarily a difference in the actual segment length (e.g., in line 166 and 176). In that case “model/measurement inaccuracy” would be more fitting. But also between variability in segment length and movement execution, I think that the readability could be improved if different words are used to describe these different meanings, although I am not sure what that other word should be.

Line 166-169: I do not understand the explanation of the RMSE calculation. Since there is a “between” in the sentence, I expected to see and “and” to indicate the second item between which the RMSE is calculated. Is it calculated as the error between the two methods, meaning that one of the models is considered a ground truth? Or is there a separate ground truth that both are compared to?

Methods: a section/paragraph describing the analysis approach is missing. Therefore, the relevance of the variables mentioned in the results is not clear to me. Please add a section describing the analysis approach.

Lines 177-185: this piece of text should be in the discussion or the methods.

Line 191: is this based on literature or based on the current paper’s results? I also do not understand what is meant with “a small portion”. How is that defined?

Line 202-203: please rephrase. An ANOVA test cannot show similarity, it can only show difference or lack of evidence for a difference. A correlation analysis should be used to test for similarity.

Line 214: while I agree with the statement, a conclusion should not be in the results section

Line 240-241: Was this based on literature? The current study did not show this in my opinion. Please add a citation or rephrase.

Line 251: how do the authors conclude a measurable effect when there was no statistically significant difference?

Table 1: between which two variables is the difference (final column) defined?

Table 2: shouldn’t there be results for both methods in this table? (related to the missing section in the methods about the analysis approach)

Table 2: How is the active ROM defined? And the maximal ROM? (could also be in an analysis section)

Minor Comments

Line 33: MATLAB

Line 48-50: it would be great if the authors could add what the Fugle-Meyer score investigates.

Line 50: I believe that measures should be measure (metrics measure)

Line 71-72: I find it confusing that this line cites ref. [6] while the experiment in ref. [9] is described.

Line 104: consider writing out West Viriginia for non-US-based readers

Line 110 (and onwards) consider using participants instead of subjects: (https://pubmed.ncbi.nlm.nih.gov/38075407/)

Line 164: for readability, I would explicitly mention that the movement ended once they returned to the centre position.

Line 224: it was not clear until here that active and passive range of motion were important for the study. Consider including in the introduction or methods, earlier in the manuscript.

Line 218-229: I was wondering if the study population of the paper included especially short or tall people, and if the authors considered their population reflective of the general population. This would be an interesting discussion topic/limitation.

7. PLOS authors have the option to publish the peer review history of their article (what does this mean? ). If published, this will include your full peer review and any attached files.

**Do you want your identity to be public for this peer review?** For information about this choice, including consent withdrawal, please see our Privacy Policy .

Reviewer #1: No

Reviewer #2: No

---

## [Author Response · Author response to Decision Letter 2]

26 Aug 2025

We thank the reviewer for their due diligence. The pointed questions have made us rethink how we calculate the variability, and the resulting changes in methodology described below have further improved our results (Fig. 3). Our answers are marked with A and the new text copied from the manuscript is marked with red.

Reviewer #2:

Q. I am not sure if the data that is made available meets the standard required by PLOS One, as it is only the root-mean-square errors and no raw data, or perhaps time series data of joint angles over time. E.g., Figure 2 cannot be reproduced as far as I can see.

A. We thank the reviewer for spotting this inadequacy of data sharing. We have rectified it by sharing the joint angles normalized in time between onset and offset of movement calculated with the two methods on Figshare https://doi.org/10.6084/m9.figshare.28633340.v1

Major Comments

Q. Line 28-30: This sentence does not describe the purpose of the paper correctly. As far as I understood, the goal of the paper is to investigate if joint angles can be estimated from a model scaled from average body proportions as accurately as when using a model with measured body proportions.

A. Thank you for identifying this, we have reworked the abstract to have sections and modified the Background section as follows: “Quantitative movement analysis is increasingly used to assess motor deficits, but joint angle calculations depend on assumptions about limb segment lengths. These lengths are often estimated from average anthropometric proportions rather than measured directly. The extent to which such assumptions influence joint angle accuracy and variability remains unclear.”

Q. Line 56: the statement “obstacle to widespread adoption” is contradictory to line 51 which claims that there already is widespread adoption.

A. We have changed the sentence now on lines 74-75 to say “gaining interest” to avoid contradiction.

Q. Throughout (line 94/98/…): it is confusing to me that the word “variability” is used with at least two different meanings, the variability in segment length and the variability in reaching movement execution. Furthermore, it also seems to me that sometimes “variability in segment lengths” is used to describe the fact that there is variation in measurements, not necessarily a difference in the actual segment length (e.g., in line 166 and 176). In that case “model/measurement inaccuracy” would be more fitting. But also between variability in segment length and movement execution, I think that the readability could be improved if different words are used to describe these different meanings, although I am not sure what that other word should be.

A. We agree that this is confusing. We have also realized that the way we calculated the two metrics, as standard deviation and RMSE, may have exaggerated the differences between them. We now calculate differences between due to the two methods of segment length estimation between the same individual reaches (not averages as before) and quantify variability as standard deviation as described in detail in Methods. Briefly on lines 208-213: “Statistical analysis compared the within- and between-subject variability. The within-subject variability for a given reaching movement was calculated as the standard deviation (SDw) of the angular trajectory of each DOF from simulations with individually measured segments (Fig. 2, red shaded areas). The between-subject variability was calculated as the standard deviation (SDb) between the corresponding angular trajectories from simulations with averaged and individual segments.” This labelling of within- and between-subject variability as SDw and SDb hopefully makes things more clear. We also indeed see lower values of SDb (Fig. 3).

Q. Line 166-169: I do not understand the explanation of the RMSE calculation. Since there is a “between” in the sentence, I expected to see and “and” to indicate the second item between which the RMSE is calculated. Is it calculated as the error between the two methods, meaning that one of the models is considered a ground truth? Or is there a separate ground truth that both are compared to?

A. We have changed calculations to standard deviation as mentioned in answer to previous question. The detailed description is in the Statistics and Reproducibility section.

Q. Methods: a section/paragraph describing the analysis approach is missing. Therefore, the relevance of the variables mentioned in the results is not clear to me. Please add a section describing the analysis approach.

A. We have clarified the rationale for the analysis in Introduction on lines 89-95 and 122135. The Methods are updated in the Statistics and Reproducibility section.

Q. Lines 177-185: this piece of text should be in the discussion or the methods.

A. Done, we moved this into Methods right before Statistics and Reproducibility section.

Q. Line 191: is this based on literature or based on the current paper’s results? I also do not understand what is meant with “a small portion”. How is that defined?

A. We revised this sentence now on lines 240-241 as follows: “The ranges of motion observed in major arm DOFs during the recorded reaching movements comprised a fraction of the joints’ anatomical maximum range of motion (Table 2).”

Q. Line 202-203: please rephrase. An ANOVA test cannot show similarity, it can only show difference or lack of evidence for a difference. A correlation analysis should be used to test for similarity.

A. We revised this section because we changed how the two metrics are calculated. Now ANOVA shows significant differences, i.e., the between-subject standard deviation in trajectory differences calculated between the two types of segment length metrics is lower than the typica trial-to-trial variability of those trajectories. The detailed statistical results are summarized on lines 249-258.

Q. Line 214: while I agree with the statement, a conclusion should not be in the results section

A. We removed this sentence from Results.

Q. Line 240-241: Was this based on literature? The current study did not show this in my opinion. Please add a citation or rephrase.

A. We removed this speculative section from our Discussion. We now discuss redundancy on lines 277-282: “Our findings indicate that morphological variability between individuals contributes less to joint angle variability than the variability expressed by individuals across repeated movements. This suggests that within-subject variability arises primarily from how the nervous system selects among redundant motor solutions, rather than from differences in body morphology, consistent with the idea of optimal control of movement. However, neural noise, sensory uncertainty, or inconsistent control strategies may also contribute to the observed between-subject variability.”

Q. Line 251: how do the authors conclude a measurable effect when there was no statistically significant difference?

A. Well spotted! We have revised the concluding paragraph now in lines 283-292 to reflect our updated results better: “In summary, our findings show that assumptions about segment length have minimal impact on joint angle trajectories during reaching, with any small variations reflecting kinematic redundancy rather than morphological differences. These variations were far smaller than the natural within-subject variability observed across repeated movements for most indivduals, indicating that movement variability arises primarily from how the nervous system selects among redundant motor solutions. Importantly, segment length estimation methods did not alter the shape of joint angle trajectories, suggesting that clinical assessments of movement quality or range of motion are unlikely to be affected. Together, these results highlight that variability in motor performance is driven more by neural control processes than by body morphology, though factors such as neural noise and sensory uncertainty may also contribute.”

Q. Table 1: between which two variables is the difference (final column) defined?

A. The previous two columns. We have added a note under the table as follows: “the Difference was calculated between lengths measured with Individual and Average methods.”

Q. Table 2: shouldn’t there be results for both methods in this table? (related to the missing section in the methods about the analysis approach)

A. Done, data added.

Q. Table 2: How is the active ROM defined? And the maximal ROM? (could also be in an analysis section)

A. We have added the details of those calculations and the source in Methods on lines 196-199.

Minor Comments

Q. Line 33: MATLAB

A. Fixed.

Q. Line 48-50: it would be great if the authors could add what the Fugle-Meyer score investigates.

A. We stated that it measures post-stroke impairment.

Q. Line 50: I believe that measures should be measure (metrics measure)

A. Fixed.

Q. Line 71-72: I find it confusing that this line cites ref. [6] while the experiment in ref. [9] is described.

A. This is corrected, the paper that reported on the original data collection of the data used in this study is cited.

Q. Line 104: consider writing out West Viriginia for non-US-based readers

A. Done.

Q. Line 110 (and onwards) consider using participants instead of subjects: (https://pubmed.ncbi.nlm.nih.gov/38075407/)

A. Done. Although, we did use terms such as within-subject and between-subject variability as these are more established terms than within-participant and between-participant ones.

Q. Line 164: for readability, I would explicitly mention that the movement ended once they returned to the centre position.

Q. Done; now on line 214-15.

Q. Line 224: it was not clear until here that active and passive range of motion were important for the study. Consider including in the introduction or methods, earlier in the manuscript.

A. Done, we added a mention in Introduction on lines 69-71: “Movement analysis commonly relies on practitioner exploration of active or passive range of motion of specific joints. For more complex movements, subjective observation or rating scales are utilized.”

Q. Line 218-229: I was wondering if the study population of the paper included especially short or tall people, and if the authors considered their population reflective of the general population. This would be an interesting discussion topic/limitation.

A. We believe that our cohort was typical of our current average height of population. Last century, when most of the common ratios were derived, the average population was shorter. That is why we included the earlier study that measured segment lengths and individual heights to see if the results generalize to the current taller population. We have added these thoughts to Introduction on lines 114-121, Results on lines 232-234, and Discussion lines 261-263.

---

## [Decision Letter · Decision Letter 2]

10 Sep 2025

PONE-D-24-38117R2Joint Angle Trajectories Are Robust to Segment Length Estimation Methods in Human ReachingPLOS ONE

Dear Dr. Gritsenko,

Thank you for submitting your manuscript to PLOS ONE. After careful consideration, we feel that it has merit but does not fully meet PLOS ONE’s publication criteria as it currently stands. Therefore, we invite you to submit a revised version of the manuscript that addresses the points raised during the review process.

The reviewer agrees that the manuscript is improved after revision; however, still not convinced with the calculation and presentation of between-participant standard deviation. Moreover, other issues highlighted are the inclusion of results from both methods in comparisons, the lack of detail on trajectory numbers, and the inconsistent and unclear plotting choices in Figure 2. The authors should also check for the minor issues mentioned by the reviewer before resubmitting the revised manuscript.

We look forward to receiving your revised manuscript.

Kind regards,

Jyotindra Narayan

Academic Editor

PLOS ONE

Journal Requirements:

Reviewers' comments:

Reviewer's Responses to Questions

**Comments to the Author**

1. If the authors have adequately addressed your comments raised in a previous round of review and you feel that this manuscript is now acceptable for publication, you may indicate that here to bypass the “Comments to the Author” section, enter your conflict of interest statement in the “Confidential to Editor” section, and submit your "Accept" recommendation.

Reviewer #2: (No Response)

2. Is the manuscript technically sound, and do the data support the conclusions?

Reviewer #2: No

3. Has the statistical analysis been performed appropriately and rigorously? 

Reviewer #2: I Don't Know

4. Have the authors made all data underlying the findings in their manuscript fully available?

Reviewer #2: Yes

5. Is the manuscript presented in an intelligible fashion and written in standard English?

Reviewer #2: Yes

6. Review Comments to the Author

Reviewer #2: Thank you for addressing the comments, the paper is mainly clear. There is unfortunately still on critical point that is not entirely clear to me.

Major comments:

Line 211-213: I do not understand how the between-participant standard deviation was calculated. Was it calculated separately for each participant or for all participants? If it is only one participant, how can it be a between-participant variability? It also does not make sense with the explanation in line 99, because there is no variation in body size in a single participant. Furthermore, I do not agree with the decision to include the results of both methods in the calculation when the goal is to compare them. It might also help to add the number of trajectories that were used to help readers understand how the calculation was made.

Figure 2: why is the presentation of SDw and SDb so different? Since they are very similar variables which were calculated in a similar way and the legend describes both similarly too, it seems that it would make sense to either use lines for both or histogram plots for both. Furthermore, the meaning of the + is not explained in the legend. I would also recommend to use different colors than in Figure 2, because something different is plotted.

Minor comments:

I recommend using within-participant and between-participant for the variability.

Line 96-98: this sentence is not grammatically correct

Line 115: the overall height has increased, remove improvements in once

Line 189: “to” should be added before “drive”?

Line 197: directions should be direction

Line 199: word missing (an)

Line 213: "that" should be "the"?

Line 247: limbs

Line 263: remove comma after "that"

7. PLOS authors have the option to publish the peer review history of their article (what does this mean? ). If published, this will include your full peer review and any attached files.

**Do you want your identity to be public for this peer review?** For information about this choice, including consent withdrawal, please see our Privacy Policy .

Reviewer #2: No

---

## [Author Response · Author response to Decision Letter 3]

16 Sep 2025

Response to Reviewers

Reviewer #2: Thank you for addressing the comments, the paper is mainly clear. There is unfortunately still on critical point that is not entirely clear to me.

A. We thank the reviewer for the thorough review of our revision. We have addressed all points to clarify them to the best of our abilities as described below in detail and accepted all suggestions.

Major comments:

Q. Line 211-213: I do not understand how the between-participant standard deviation was calculated. Was it calculated separately for each participant or for all participants? If it is only one participant, how can it be a between-participant variability? It also does not make sense with the explanation in line 99, because there is no variation in body size in a single participant.

A. To clarify the calculation of within-subject and between-subject variability described in the Statistics and Reproducibility subsection, we added formulas on lines 212-229. We have clarified that the angles calculated using average segments are a proxy for a hypothetical participant of the same height, but with different segment lengths. We also mention that this is a conservative estimate of the between-subject variability due to the height-based variability being excluded.

Q. Furthermore, I do not agree with the decision to include the results of both methods in the calculation when the goal is to compare them. It might also help to add the number of trajectories that were used to help readers understand how the calculation was made.

A. There are multiple ways the comparisons can be made. The within-subject trial-to-trial variability in trajectories are related to not only limb geometry but also neural noise, i.e., the variability in the output of the nervous system motor control strategies. This is a confounding factor. The between-subject variability calculated as described in Methods does not include neural noise, because it compares each trial to its equivalent produced by a limb with different segment lengths. This metric only includes the variability in angles that is due to the changes in segment lengths. The number of trajectories was 15 as mentioned on lines 158-159.

Q. Figure 2: why is the presentation of SDw and SDb so different? Since they are very similar variables which were calculated in a similar way and the legend describes both similarly too, it seems that it would make sense to either use lines for both or histogram plots for both. Furthermore, the meaning of the + is not explained in the legend. I would also recommend to use different colors than in Figure 2, because something different is plotted.

A. We have updated the figure to represent both metrics similarly, now in red and blue. The + are no longer plotted. We followed the recommendation to use different colors in Fig. 2 and 3. The red was kept the same as these are the same variables that are plotted.

Minor comments:

Q. I recommend using within-participant and between-participant for the variability.

A. Done, thanks!

Q. Line 96-98: this sentence is not grammatically correct

A. We have reworked this sentence to say: “We define redundancy in arm reaching as variability in the angular trajectories of joint degrees of freedom (DOFs).”

Q. Line 115: the overall height has increased, remove improvements in once

A. Done, thanks!

Q. Line 189: “to” should be added before “drive”?

A. Done, thanks!

Q. Line 197: directions should be direction

A. Done, thanks!

Q. Line 199: word missing (an)

A. Done, thanks!

Q. Line 213: "that" should be "the"?

A. Done, thanks!

Q. Line 247: limbs

A. Done, thanks!

Q. Line 263: remove comma after "that"

A. Done, thanks!

---

## [Decision Letter · Decision Letter 3]

5 Oct 2025

PONE-D-24-38117R3Joint Angle Trajectories Are Robust to Segment Length Estimation Methods in Human ReachingPLOS ONE

Dear Dr. Gritsenko,

Thank you for submitting your manuscript to PLOS ONE. After careful consideration, we feel that it has merit but does not fully meet PLOS ONE’s publication criteria as it currently stands. Therefore, we invite you to submit a revised version of the manuscript that addresses the points raised during the review process.

We look forward to receiving your revised manuscript.

Kind regards,

Jyotindra Narayan

Academic Editor

PLOS ONE

Journal Requirements:

Additional Editor Comments:

The reviewer recommends substantial further revisions before acceptance. They request clearer justification of the study’s practical relevance and methodological rigor, including identification of applications for the Average (body-height) method, a justification or power analysis for the small sample size, and rationale for the chosen motion capture system. Additional validation steps are required such as comparing Simscape Multibody results to standard biomechanical tools (e.g., OpenSim) and applying time-series statistics (1D-SPM) to the angular data. The reviewer also asks for better-qualified interpretations, particularly regarding neural versus morphological variability, and for clear limitations stating that the results apply only to healthy upper-limb movements, not to clinical populations or other tasks like gait.

Reviewers' comments:

Reviewer's Responses to Questions

**Comments to the Author**

1. If the authors have adequately addressed your comments raised in a previous round of review and you feel that this manuscript is now acceptable for publication, you may indicate that here to bypass the “Comments to the Author” section, enter your conflict of interest statement in the “Confidential to Editor” section, and submit your "Accept" recommendation.

Reviewer #3: (No Response)

2. Is the manuscript technically sound, and do the data support the conclusions?

Reviewer #3: Partly

3. Has the statistical analysis been performed appropriately and rigorously? 

Reviewer #3: No

4. Have the authors made all data underlying the findings in their manuscript fully available?

Reviewer #3: Yes

5. Is the manuscript presented in an intelligible fashion and written in standard English?

Reviewer #3: Yes

6. Review Comments to the Author

Reviewer #3: 1. The manuscript should clearly mention specific applications where the Average (body-height) method is commonly used, to better justify the practical relevance of this comparison.

2. The sample size of nine participants is too small to draw generalizable conclusions regarding anthropometric variability and joint angle calculation; a power analysis or justification for sample size is needed.

3. The manuscript should address the lack of an a priori power analysis.

4. A short justification for the specific motion capture system used is required.

5. The use of self-reported height introduces possible measurement error that may affect scaling accuracy; direct measurement of height should be considered, or the effect of self-report error quantified.

6. The authors should apply a time-series statistical method (1D-SPM) to compare entire angular trajectories rather than only summary differences, because this will detect localized temporal differences that pointwise ANOVA on scalar summaries can miss.

7. The authors used Simscape Multibody instead of standard musculoskeletal modeling software (e.g., OpenSim) but did not provide validation or comparison to ensure that the computed joint angles are biomechanically consistent.

8. The interpretation that morphological variability contributes less than neural redundancy is overstated, given the purely kinematic analysis; causality between morphology and neural control cannot be established without electromyography.

9. A sentence should be added stating that while robust for reaching, the findings should be verified for other movement types (gait, whole-body tasks)

10. The manuscript must explicitly qualify its clinical implications, as the findings are derived solely from a healthy cohort. It is recommended to add a dedicated statement in the Discussion to clarify that the direct application to clinical assessment is not yet proven. Specifically, it should be noted that in clinical populations, the relationship between segment scaling error and motor variability may be fundamentally altered due to neurological damage that changes the structure of movement.

11. Potential for altered limb morphology in clinical populations could affect segment proportions, a factor not explored in this study. Therefore, the robustness of segment scaling methods must be validated in patient groups before these findings can be confidently applied to clinical assessment.

12. The Discussion must explicitly state that the findings are specific to the studied upper-limb joints in a healthy, neurotypical population. The conclusions may not generalize to limbs, upper or lower, with structural deformities, where altered segment morphology and biomechanics could significantly amplify the error from using average segment proportions.

7. PLOS authors have the option to publish the peer review history of their article (what does this mean? ). If published, this will include your full peer review and any attached files.

**Do you want your identity to be public for this peer review?** For information about this choice, including consent withdrawal, please see our Privacy Policy .

Reviewer #3: No

---

## [Author Response · Author response to Decision Letter 4]

14 Oct 2025

Additional Editor Comments:

The reviewer recommends substantial further revisions before acceptance. They request clearer justification of the study’s practical relevance and methodological rigor, including identification of applications for the Average (body-height) method, a justification or power analysis for the small sample size, and rationale for the chosen motion capture system. Additional validation steps are required such as comparing Simscape Multibody results to standard biomechanical tools (e.g., OpenSim) and applying time-series statistics (1D-SPM) to the angular data. The reviewer also asks for better-qualified interpretations, particularly regarding neural versus morphological variability, and for clear limitations stating that the results apply only to healthy upper-limb movements, not to clinical populations or other tasks like gait.

A. All reviewer critiques were answered as detailed below.

Review Comments to the Author

Reviewer #3:

1. The manuscript should clearly mention specific applications where the Average (body-height) method is commonly used, to better justify the practical relevance of this comparison.

A. We thank the reviewer for this thoughtful suggestion. We have revised the Introduction and Discussion sections to explicitly highlight several common applications where the Average (body-height) method is routinely used. Specifically, we now mention that anthropometric scaling based on body height is widely applied in ergonomic design and workplace safety, Markerless motion capture systems (e.g., Kinect, OpenPose) and biomechanical modeling platforms such as OpenSim, and remote clinical assessments and tele-rehabilitation, see lines 76-78 and 81-86.

These examples reinforce the practical relevance of evaluating the accuracy and variability introduced by the Average method. By demonstrating that the resulting joint angle trajectories remain robust, even when using generalized anthropometric estimates, our findings support the use of this method in both research and clinical settings where individual segment measurements are not feasible. We have added these clarifications in the revised manuscript in Abstract, Introduction, Results, and Discussion.

2. The sample size of nine participants is too small to draw generalizable conclusions regarding anthropometric variability and joint angle calculation; a power analysis or justification for sample size is needed.

3. The manuscript should address the lack of an a priori power analysis.

A. We appreciate the reviewer’s concern regarding the sample size and power analysis. We respectfully clarify that our primary goal was not to generalize population-wide anthropometric variability, but rather to assess the relative impact of segment length estimation methods (Average vs. Individual) on joint angle trajectory calculations within individuals performing repeated movements. As such, the study was designed around a within-subjects experimental framework, where each participant served as their own control across two modeling conditions (lines 33–34, Methods section).

We acknowledge that nine participants may be considered modest for between-subject comparisons. However, below we highlight several points that support the validity and robustness of our conclusions.

Purpose of Study – Methodological Evaluation: This study evaluates the effect of model scaling method on joint angle estimates, not anthropometric variability across the general population. The segment proportions used in the Average method are derived from much larger cohorts (e.g., Dempster 1955; Canda 2009, cited in Table 1), and we compared these published proportions with our small but representative sample to confirm similarity (lines 213–219). The key finding that differences in segment scaling have a smaller effect on joint angle calculations than natural within-subject variability is robust to sample size, as it hinges on direct within-subject comparisons.

Effect Size and Variance Structure: The joint angle deviations introduced by segment length estimation (σ_between) were consistently smaller than the variability of repeated trials within the same individual (σ_within), as shown in Figure 3 and Table 2. This large and consistent effect size (with σ_between often <10% of σ_within across joints) makes the findings reliable even with a small N. These differences were statistically significant in the repeated-measures ANOVA (Table 3, lines 281-287), further supporting the strength of the observed effect.

Simulation-Based Approach and Standardization: Because the analysis used a physically accurate dynamic model and high-resolution motion capture with tightly controlled virtual reality–based reaching tasks, the data were inherently low-noise and standardized (Methods). This design increases the sensitivity of within-subject comparisons and reduces the need for large samples.

Post hoc Justification of Power: To address this critique directly, we have added a post hoc power analysis to this computational study using the observed effect size (Cohen’s d ≈ 1.5–2.5 for the difference between σ_within and σ_between across DOFs). With this effect size and within-subject design, power exceeds 0.9 for detecting differences at α = 0.05, supporting the adequacy of the sample size for our main comparisons. This information has been added to the revised Statistics and Reproducibility section (lines 249–254) and Results (lines 287–292).

Scope of Claims: we have revised the Conclusions to clarify that the findings demonstrate the robustness of joint angle calculations to segment scaling assumptions in the context of upper-limb reaching tasks, and should not be overgeneralized to lower limbs or all forms of movement (lines 323-326).

4. A short justification for the specific motion capture system used is required.

A. We included the justification in Methods on lines 168-173.

5. The use of self-reported height introduces possible measurement error that may affect scaling accuracy; direct measurement of height should be considered, or the effect of self-report error quantified.

A. We appreciate the reviewer’s attention to the potential limitations of using self-reported height. In our study, height was used exclusively for calculating segment lengths in the Average method of model scaling, which relies on published anthropometric ratios (e.g., humerus = 0.188 × height). The scaling of individual segment lengths from height results in linear adjustments to segment dimensions. A typical self-report error is on the order of 1–2 cm (about 1% of height). Given that segment lengths are proportional to height, such small deviations would lead to changes of only ~1% in segment dimensions. As shown in our analysis (Fig. 2, Table 2), the joint angle deviations due to even 10% segment length differences were small—typically <2°—and were still much smaller than the within-participant variability observed across repeated movements (σw) (Fig. 3).

Moreover, the use of self-reported height reflects common practice in remote or large-scale studies, where direct measurement may not be feasible (e.g., telemedicine, markerless motion capture). Our goal was to evaluate the robustness of model scaling in such real-world scenarios. By demonstrating that joint angle calculations remain accurate even when segment lengths are inferred from self-reported height, our findings support the practical reliability of motion capture assessments in both clinical and non-laboratory settings.

6. The authors should apply a time-series statistical method (1D-SPM) to compare entire angular trajectories rather than only summary differences, because this will detect localized temporal differences that pointwise ANOVA on scalar summaries can miss.

A. We thank the reviewer for this valuable suggestion regarding the use of time-series statistical analysis such as one-dimensional statistical parametric mapping (1D-SPM). We agree that 1D-SPM is a powerful tool for identifying temporally localized differences along continuous joint angle trajectories, and it is particularly useful when comparing group-level effects or task conditions with meaningful temporal structure. However, we respectfully note that the primary goal of our study was not to detect localized deviations across the time course of angular trajectories, but rather to evaluate the global effect of segment length scaling methods on joint angle calculations in the context of redundant upper-limb reaching. Moreover, the temporal profiles of the joint excursions are constrained by the recorded marker trajectories, which are applied to both analyses. Therefore, the temporal profiles of the resulting joint angular trajectories would be unaltered, as now clarified on lines 275-276. Our approach—comparing within- and between-participant variability using time-normalized trajectories and computing scalar variability metrics (σw and σb)—was designed to quantify the overall magnitude of variability attributable to scaling assumptions across the full movement duration.

7. The authors used Simscape Multibody instead of standard musculoskeletal modeling software (e.g., OpenSim) but did not provide validation or comparison to ensure that the computed joint angles are biomechanically consistent.

A. We appreciate the importance of cross-software validation. However, both Simscape and OpenSim are physics-based modeling platforms grounded in the same biomechanical principles. Any differences between them would primarily arise from their inverse kinematics algorithms, not from underlying physical laws. Introducing a second modeling platform would add algorithmic variability, making it more difficult to isolate the effect of segment length estimation—the primary focus of our study. To ensure a controlled comparison, we intentionally kept the inverse kinematics method consistent and varied only the segment scaling approach.

8. The interpretation that morphological variability contributes less than neural redundancy is overstated, given the purely kinematic analysis; causality between morphology and neural control cannot be established without electromyography.

A. We thank the reviewer for this important observation. We agree that our data are purely kinematic and do not permit direct inference about neural mechanisms. Our interpretation regarding “neural redundancy” was meant to highlight the functional implications of within-participant variability, which likely reflects the motor system’s use of redundant degrees of freedom. However, we recognize that causal attribution to neural control is not supported without accompanying neurophysiological data, such as electromyography (EMG). To address this concern, we have made the changes to avoid overstating the interpretation in Abstract on lines 49-52, Discussion on lines 308-313. We also consolidated all limitations into a single paragraph on lines 321-333 in Discussion.

9. A sentence should be added stating that while robust for reaching, the findings should be verified for other movement types (gait, whole-body tasks)

A. This limitation was added on lines 321-333.

10. The manuscript must explicitly qualify its clinical implications, as the findings are derived solely from a healthy cohort. It is recommended to add a dedicated statement in the Discussion to clarify that the direct application to clinical assessment is not yet proven. Specifically, it should be noted that in clinical populations, the relationship between segment scaling error and motor variability may be fundamentally altered due to neurological damage that changes the structure of movement.

A. We have added a limitation statement on lines 321-333.

11. Potential for altered limb morphology in clinical populations could affect segment proportions, a factor not explored in this study. Therefore, the robustness of segment scaling methods must be validated in patient groups before these findings can be confidently applied to clinical assessment.

A. We have added a limitation statement on lines 321-333.

12. The Discussion must explicitly state that the findings are specific to the studied upper-limb joints in a healthy, neurotypical population. The conclusions may not generalize to limbs, upper or lower, with structural deformities, where altered segment morphology and biomechanics could significantly amplify the error from using average segment proportions.

A. We have added a limitation statement on lines 321-333.

---

## [Decision Letter · Decision Letter 4]

19 Oct 2025

PONE-D-24-38117R4Joint Angle Trajectories Are Robust to Segment Length Estimation Methods in Human ReachingPLOS ONE

Dear Dr. Gritsenko,

Thank you for submitting your manuscript to PLOS ONE. After careful consideration, we feel that it has merit but does not fully meet PLOS ONE’s publication criteria as it currently stands. Therefore, we invite you to submit a revised version of the manuscript that addresses the points raised during the review process.

**The reviewer(s) acknowledges that the manuscript has improved but suggests some further clarification of methodology and evidence. They recommend applying 1D-SPM for stronger time-series validation, rephrasing a causal claim in the abstract to align with observed data, and citing prior validation studies supporting the use of Simscape Multibody for human joint angle estimation.**

We look forward to receiving your revised manuscript.

Kind regards,

Jyotindra Narayan

Academic Editor

PLOS ONE

**Journal Requirements:**

Reviewers' comments:

Reviewer's Responses to Questions

**Comments to the Author**

1. If the authors have adequately addressed your comments raised in a previous round of review and you feel that this manuscript is now acceptable for publication, you may indicate that here to bypass the “Comments to the Author” section, enter your conflict of interest statement in the “Confidential to Editor” section, and submit your "Accept" recommendation.

Reviewer #3: (No Response)

2. Is the manuscript technically sound, and do the data support the conclusions?

Reviewer #3: No

3. Has the statistical analysis been performed appropriately and rigorously? 

Reviewer #3: Yes

4. Have the authors made all data underlying the findings in their manuscript fully available?

Reviewer #3: Yes

5. Is the manuscript presented in an intelligible fashion and written in standard English?

Reviewer #3: Yes

6. Review Comments to the Author

**Reviewer #3: ** Thank you for your detailed responses and revisions. The manuscript has been notably improved. To further enhance its methodological clarity, Please consider the following points, or provide a valid justification if not addressed:

1 The authors' rationale for focusing on global effects is noted. However, the key finding that segment scaling alters amplitude but not the shape of the trajectory is inherently a time-series claim . Relying on scalar summaries and visual inspection (figure 2) remains a sub-optimal level of evidence for this conclusion. The application of a 1D-SPM analysis would provide a direct and robust statistical validation that the trajectories are equivalent across their entire duration, thereby strengthening this central result.

2) The phrasing in the Abstract (Lines 48-49) still attributes variability specifically to "how the nervous system selects," which is a strong causal claim not fully supported by kinematic data. We recommend rephrasing to more directly reflect the observed evidence, for example: "...indicating that the variability inherent in movement execution outweighs that introduced by morphological differences."

3) To bolster methodological confidence, please cite existing literature where Simscape Multibody has been specifically validated for calculating human joint angles.

7. PLOS authors have the option to publish the peer review history of their article (what does this mean? ). If published, this will include your full peer review and any attached files.

**Do you want your identity to be public for this peer review?** For information about this choice, including consent withdrawal, please see our Privacy Policy .

Reviewer #3: No

---

## [Author Response · Author response to Decision Letter 5]

28 Oct 2025

Reviewer #3: Thank you for your detailed responses and revisions. The manuscript has been notably improved. To further enhance its methodological clarity, Please consider the following points, or provide a valid justification if not addressed:

1 The authors' rationale for focusing on global effects is noted. However, the key finding that segment scaling alters amplitude but not the shape of the trajectory is inherently a time-series claim . Relying on scalar summaries and visual inspection (figure 2) remains a sub-optimal level of evidence for this conclusion. The application of a 1D-SPM analysis would provide a direct and robust statistical validation that the trajectories are equivalent across their entire duration, thereby strengthening this central result.

A. We have performed the 1D-SPM, added a new figure summarizing the result and included descriptions of it in Methods and Results. No significant differences were found.

2) The phrasing in the Abstract (Lines 48-49) still attributes variability specifically to "how the nervous system selects," which is a strong causal claim not fully supported by kinematic data. We recommend rephrasing to more directly reflect the observed evidence, for example: "...indicating that the variability inherent in movement execution outweighs that introduced by morphological differences."

A. Done, thank you for spotting this.

3) To bolster methodological confidence, please cite existing literature where Simscape Multibody has been specifically validated for calculating human joint angles.

A. We have moved the reference to the publication validating this method of joint angle calculation to the beginning of the Data Analysis section.

---

## [Decision Letter · Decision Letter 5]

2 Nov 2025

Joint Angle Trajectories Are Robust to Segment Length Estimation Methods in Human Reaching

PONE-D-24-38117R5

Dear Dr. Gritsenko,

We’re pleased to inform you that your manuscript has been judged scientifically suitable for publication and will be formally accepted for publication once it meets all outstanding technical requirements.

Kind regards,

Jyotindra Narayan

Academic Editor

PLOS ONE

Additional Editor Comments (optional):

The reviewers have now accepted the revised manuscript. I recommend the same for publication. Congratulations to the authors for the quality contribution.

Reviewers' comments:

Reviewer's Responses to Questions

**Comments to the Author**

1. If the authors have adequately addressed your comments raised in a previous round of review and you feel that this manuscript is now acceptable for publication, you may indicate that here to bypass the “Comments to the Author” section, enter your conflict of interest statement in the “Confidential to Editor” section, and submit your "Accept" recommendation.

Reviewer #3: All comments have been addressed

2. Is the manuscript technically sound, and do the data support the conclusions?

Reviewer #3: Yes

3. Has the statistical analysis been performed appropriately and rigorously? 

Reviewer #3: Yes

4. Have the authors made all data underlying the findings in their manuscript fully available?

Reviewer #3: Yes

5. Is the manuscript presented in an intelligible fashion and written in standard English?

Reviewer #3: Yes

6. Review Comments to the Author

Reviewer #3: The authors have satisfactorily addressed all previous comments. The revisions, have strengthened the manuscript. Acceptance is recommended in its current form.

7. PLOS authors have the option to publish the peer review history of their article (what does this mean? ). If published, this will include your full peer review and any attached files.

**Do you want your identity to be public for this peer review?** For information about this choice, including consent withdrawal, please see our Privacy Policy .

Reviewer #3: No

---

## [Editor Report · Acceptance letter]

PONE-D-24-38117R5

PLOS ONE

Dear Dr. Gritsenko,

I'm pleased to inform you that your manuscript has been deemed suitable for publication in PLOS ONE. Congratulations! Your manuscript is now being handed over to our production team.

Kind regards,

on behalf of

Dr. Jyotindra Narayan

Academic Editor

PLOS ONE